# Non-Linear Spectral Dimensionality Reduction Under Uncertainty

## Abstract

In this paper, we consider the problem of non-linear dimensionality reduction under uncertainty, both from theoretical and algorithmic perspectives. Since real-world data usually contain measurements with uncertainties and artifacts, the input space in the proposed framework consists of probability distributions to model the uncertainties associated with each sample. We propose a new dimensionality reduction framework, called Non-linear Graph Embedding with Data Uncertainty (NGEU), which leverages uncertainty information and extends the Graph Embedding (GE) framework. It can be used to extend several traditional approaches, such as KPCA, and MDA/KMFA, encapsulated in the GE framework to take as inputs the probability distributions instead of the original data. We show that the proposed NGEU formulation exhibits a global closed-form solution, and we analyze, based on the Rademacher complexity, how the underlying uncertainties theoretically affect the generalization ability of the framework. Empirical results on different datasets show the effectiveness of the proposed framework.

## 1 Introduction

Uncertainty refers to situations including imperfect or unknown data (Li et al., 2012; Ovadia et al., 2019). Data are usually obtained through sensors, e.g., camera, accelerometer, or microphone, and, thus, can be exposed to measurement inaccuracies and noise. Although total elimination of uncertainty is impractical, modeling and attenuating the effect of noise and uncertainty is critical for trust-worthy applications, such as decision-making systems. In particular, a robust machine learning method needs to take into consideration this type of partly inaccurate data and exhibit some degree of robustness with respect to such errors.

Recently, modeling of uncertainty has gained attention in machine learning community in general (Kirsch et al., 2019; Zhang et al., 2019; Bhadra et al., 2010) and in dimensionality reduction (DR) in particular (Gao et al., 2018; Saeidi et al., 2015; Gajamannage et al., 2019; Laakom et al., 2022). Various DR techniques have been proposed which consider data uncertainty and inaccuracies (Wang et al., 2015a; Li et al., 2020; Gerber et al., 2007; Lourenço et al., 2017). We note two main approaches for dealing with uncertainty in DR. In the first one, called sample-wise uncertainty, the unreliability is assumed to occur at the sample level. The key hypothesis of these approaches is: 'Some available samples are out-of-distribution and are more likely to be outliers'. Thus, they need to be discarded and not used to learn the low-dimensional embedding (Pham, 2018). Various DR methods have been extended based on this assumption, for examples the approaches in (Kong & Ding, 2014; Luo et al., 2011; Chumachenko et al., 2021) for Linear Discriminant Analysis (LDA) (Schalkoff, 2007) and the approaches in (Nie et al., 2014; Zhang et al., 2015; Wang et al., 2015b; Park & Choi, 2009) for Principal Component Analysis (PCA) (Wold et al., 1987).

In the second type of exploiting uncertainty, called feature-wise uncertainty, the unreliability is assumed on the feature level. The main hypothesis of this paradigm is that certain data features are corrupted by noise (Saeidi et al., 2015). In (Laakom et al., 2022), several traditional linear DR techniques, e.g., LDA and Marginal Fisher Analysis (MFA), were extended to consider uncertainty. However, the main drawback of that approach is that uncertainty modeling is restricted to the Gaussian case, which is impractical for many applications. Moreover, it is unable to model nonlinear patterns in the input data and considers only linear

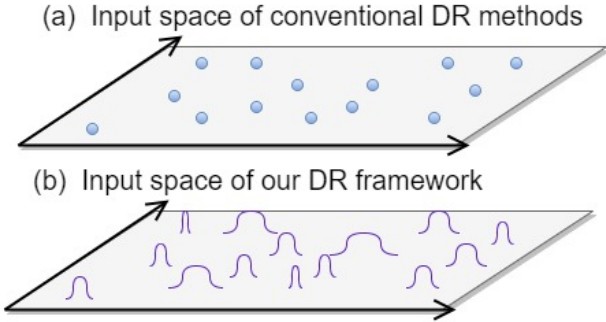

Figure 1: 2-D Illustrative comparison of the input space for (a) conventional dimensionality reduction methods and (b) our framework in the presence of the uncertainty information. Blue circles denote the conventional samples. Blue curves model the input data in our framework, i.e., normal distributions in this case.

data projections, whereas in the case of non-linearly distributed data, a meaningful embedding cannot be achieved by a linear projection.

We propose NGEU, a novel framework for non-linear dimensionality reduction. It is designed as an extension of GE and leverages the available input data uncertainty information. As illustrated in Figure 1, we view our samples as probability distributions modeling the inaccuracies and uncertainties of the data, i.e., each data point has a corresponding distribution modeling the uncertainty of its features. Furthermore, unlike (Laakom et al., 2022), NGEU is not restricted to Gaussian noise and is suitable to learn from a wide range of distributions[1]. This allows for a more flexible modeling of the uncertainty in the input space. The proposed framework can be used to derive robust extensions of DR methods formulated with the kernelization of the original GE framework (Yan et al., 2007), including Kernel Principal Component Analysis (KPCA), Kernel Discriminant Analysis (KDA), and Kernel Marginal Fisher Analysis (KMFA). While it has been shown empirically that leveraging the uncertainty information via distributional data in dimensionality reduction setup improves the robustness of the model (Laakom et al., 2022), no theoretical guarantees have been provided yet. To bridge this gap, in this paper, we theoretically analyze how incorporating the uncertainty information affects the generalization ability of the methods, based on the Rademacher complexity. We show that considering uncertainty indeed yields more robust performance compared to the original GE-based methods.

The main contributions of the paper can be summarized as follows:

- We propose a novel non-linear dimensionality reduction (DR) framework that considers uncertainties in the input data.

- We reconstruct the Graph Embedding (GE) framework and provide a solution for leveraging data uncertainties in several DR approaches framed under the kernel GE framework.

- We prove that our framework has one global optimum closed-form solution.

- Based on Rademacher complexity, we provide theoretical learning guarantees for our framework and analyze the effect of uncertainty on the generalization ability of the DR methods.

## 2  A Review of Graph Embedding Framework for Dimensionality Reduction

Here, we describe the GE framework (Yan et al., 2007; Iosifidis et al., 2016) that exploits geometric data relationships to learn a low-dimensional mapping of the samples. Different dimensionality reduction techniques, including PCA, LDA, and MFA, can be formulated in this framework by using a specific setting of

---

[1]Our framework is able to incorporate any type distribution, for which equation 8 can be computed.

two graphs, called intrinsic and penalty graphs, modeling the pairwise connection between the samples (Yan et al., 2007; Miao et al., 2022; Gou et al., 2020; Örnek & Vural, 2019; Pan et al., 2009; Örnek & Vural, 2019).

Formally, let us consider the supervised DR problem and denote by $\{\mathbf{x}_i, c_i\}_{i=1}^N$ a set of D-dimensional vectors $\mathbf{x}_i \in \mathbb{R}^D$ and their corresponding class labels. We wish to learn a mapping that transforms each $\mathbf{x}_i$ into $\mathbf{y}_i \in \mathbb{R}^d$, such that $d < D$. GE assumes that the training data form the vertex set $\mathcal{V} = \{v_i\}_{i=1}^N$, where each vertex $v_i$ corresponds to a sample $\mathbf{x}_i$. Pair-wise connections between the vertices are modeled by the edges of two undirected graphs defined on $\mathcal{V}$, i.e., the intrinsic graph $\mathcal{G}(\mathcal{V}, \mathcal{E})$ modeling pair-wise data relationships to be highlighted in the edge set $\mathcal{E} = \{e_{ij}\}$ and the penalty graph $\mathcal{G}^p(\mathcal{V}, \mathcal{E}^P)$ modeling pair-wise data relationships to be suppressed in the edge set $\mathcal{E}^p = \{e_{ij}^p\}$. The strengths of the pair-wise vertex connections in $\mathcal{G}$ and $\mathcal{G}^p$ are expressed by the graph weight matrices $\mathbf{W} \in \mathbb{R}^{N \times N}$ and $\mathbf{W}^p \in \mathbb{R}^{N \times N}$, respectively. Using $\mathbf{W}$ and $\mathbf{W}^p$, the graph degree matrices $\mathbf{D} = \text{diag}(\mathbf{W1})$ and $\mathbf{D}^p = \text{diag}(\mathbf{W}^p\mathbf{1})$ and the graph Laplacian matrices $\mathbf{L} = \mathbf{D} - \mathbf{W}$ and $\mathbf{L}^p = \mathbf{D}^p - \mathbf{W}^p$ are defined, where $\mathbf{1} = [1, \dots, 1]^T \in \mathbb{R}^N$.

The 1-D mappings of $\{\mathbf{x}_i\}_{i=1}^N$ are obtained through the graph preserving criterion expressed as follows:

$$\mathbf{y}^* = \arg\min_{\mathbf{y}} \ \sum_{i \neq j} (y_i - y_j)^2 \mathbf{W}_{ij},$$
$$\text{s.t.} \quad \mathbf{y}^T \mathbf{B} \mathbf{y} = m, \tag{1}$$

where $\mathbf{B}$ can model a graph constraint, e.g., $\mathbf{B} = \mathbf{I}$, or the graph Laplacian of the penalty graph, i.e., $\mathbf{B} = \mathbf{L}^p$, $m$ is a constant, $y_i$ is the 1-D mapping of $\mathbf{x}_i$, and $\mathbf{y} \in \mathbb{R}^N$ is a vector containing the mappings for all $\mathbf{x}_i$.

**Linear case**

The 1-dimensional mappings of the vertices can be obtained using a linear projection, i.e., $\mathbf{y} = \mathbf{X}^T \mathbf{p}^*$. The objective function in (1) can be rewritten using the projection vector $\mathbf{p}^*$ as follows:

$$\mathbf{p}^* = \arg\min_{\mathbf{p}} \ \mathbf{p}^T \mathbf{X} \mathbf{L} \mathbf{X}^T \mathbf{p},$$
$$\text{s.t.} \quad \mathbf{p}^T \mathbf{X} \mathbf{B} \mathbf{X}^T \mathbf{p} = m. \tag{2}$$

The minimization problem in (2) is equivalent to the generalized eigenvalue decomposition problem

$$\left( \mathbf{X} \mathbf{L} \mathbf{X}^T \right) \mathbf{p} = \rho \left( \mathbf{X} \mathbf{B} \mathbf{X}^T \right) \mathbf{p}. \tag{3}$$

The solution vector $\mathbf{p}^*$ corresponds to the eigenvector with the minimal positive eigenvalue $\rho$. In the case where $d > 1$, the solution $\mathbf{P} \in \mathbb{R}^{D \times d}$ is constructed using the $d$ eigenvectors with the smallest positive eigenvalues.

**Non-linear case**

In the case of non-linearly distributed data, a meaningful embedding cannot be achieved by a linear projection. The kernel trick (Schölkopf, 2001; Hofmann, 2006) was employed to extend GE to nonlinear cases. Let us denote by $\phi : \mathbf{x} \hookrightarrow \mathcal{F}$ a mapping transforming $\mathbf{x}$ to a higher-dimensional Hilbert space $\mathcal{F}$ and by $\mathbf{\Phi} = [\phi(\mathbf{x}_1), \dots, \phi(\mathbf{x}_N)]$ the matrix containing the representations of the entire data set in $\mathcal{F}$. Subsequently, the data in $\mathcal{F}$ is linearly mapped to the final 1-D representation as $\mathbf{y} = \mathbf{\Phi}^T \mathbf{p}^*$. Using the Representer theorem (Dinuzzo & Schölkopf, 2012), the solution $\mathbf{p}^*$ is expressed as a linear sum of the samples in $\mathcal{F}$, i.e., $\mathbf{p}^* = \sum_j \alpha_j \phi(\mathbf{x}_j) = \mathbf{\Phi}\boldsymbol{\alpha}^*$, where $\boldsymbol{\alpha}^* = [\alpha_1, \dots, \alpha_N]^T$. The final representation can be now given as $\mathbf{y} = \mathbf{\Phi}^T \mathbf{p}^* = \mathbf{\Phi}^T \mathbf{\Phi} \boldsymbol{\alpha}^* = \mathbf{K}^T \boldsymbol{\alpha}^*$, where $\mathbf{K}$ is the so-called kernel matrix, whose element $\mathbf{K}_{ij}$ computes the inner product of a data pair $\mathbf{K}_{ij} = \phi(\mathbf{x}_i)^T \phi(\mathbf{x}_j)$. The kernel trick allows formulating many methods using the kernel matrix $\mathbf{K}$ without the need of computing the possibly infinite-dimensional representations in $\mathbf{\Phi}$. The main objective function in (1) can be now formulated as follows:

$$\boldsymbol{\alpha}^* = \arg\min_{\boldsymbol{\alpha}} \ \boldsymbol{\alpha}^T \mathbf{K} \mathbf{L} \mathbf{K}^T \boldsymbol{\alpha},$$
$$\text{s.t.} \quad \boldsymbol{\alpha}^T \mathbf{K} \mathbf{L}^p \mathbf{K}^T \boldsymbol{\alpha} = m \quad \text{or} \quad \boldsymbol{\alpha}^T \mathbf{K} \boldsymbol{\alpha} = m, \tag{4}$$

which is equivalent to

$$\left(\mathbf{KLK}^T\right)\boldsymbol{\alpha} = \rho\left(\mathbf{KBK}^T\right)\boldsymbol{\alpha}, \tag{5}$$

where $\mathbf{B}$ is equal to $\mathbf{KL}^p\mathbf{K}^T$ or $\mathbf{K}$. As in the linear case, the projection vector $\boldsymbol{\alpha}^*$ corresponds to the eigenvector with the minimal positive eigenvalue $\rho$ and for $d > 1$, the solution $\mathbf{A} \in \mathbb{R}^{N \times d}$ is constructed using the $d$ eigenvectors with the smallest positive eigenvalues.

## 3 Kernel Mean Embedding

In this section, we introduce Kernel Mean Embedding (KME) (Muandet et al., 2017; Hsu et al., 2018), because it will be used within our method. KME extends the classical kernel approach to probability distributions. Specifically, choosing a kernel implies an implicit feature map $\phi$ that represents a probability distribution $\mathbb{P}$ as a mean function. Let $\mathcal{H}$ denote the reproducing kernel Hilbert space (RKHS) of real-valued functions $f : \chi \hookrightarrow \mathbb{R}$ with the reproducing kernel $k : \chi \times \chi \hookrightarrow \mathbb{R}$. The kernel mean map $\mu : \mathfrak{B}_\chi \hookrightarrow \mathcal{H}$ is defined as

$$\mu(\mathbb{P}) : \mathbb{P} \hookrightarrow \int_\chi k(\mathbf{x}, \cdot)\, d\mathbb{P}(\mathbf{x}) = \int_\chi \phi(\mathbf{x})\, d\mathbb{P}(\mathbf{x}). \tag{6}$$

For any $\mathbb{P} \in \mathfrak{B}_\chi$, the following reproducing property holds

$$\mathbb{E}_\mathbb{P}[f] = <\mu(\mathbb{P}), f>_\mathcal{H} = \int f(\mathbf{x}) d\mathbb{P}(\mathbf{x}), \tag{7}$$

where $f$ is a function in $\mathcal{H}$ and $< \cdot, \cdot >_\mathcal{H}$ denotes the inner product in $\mathcal{H}$. Thus, we can see $\mu(\mathbb{P})$ as the feature map associated with the kernel $\mathcal{K} : \mathfrak{B}_\chi \times \mathfrak{B}_\chi \hookrightarrow \mathbb{R}$, which can be defined as follows:

$$\mathcal{K}(\mathbb{P}, \mathbb{Q}) = <\mu(\mathbb{P}), \mu(\mathbb{Q})>_\mathcal{H} = \int\int k(\mathbf{x}, \mathbf{z}) d\mathbb{P}(\mathbf{x}) d\mathbb{Q}(\mathbf{z}). \tag{8}$$

$k(\cdot, \cdot)$ is usually referred to as first level kernel and $\mathcal{K}(\cdot, \cdot)$ as second level kernel. Equation (8) implies that the explicit expression of $\mu(\cdot)$ is not needed but only the inner products. This property resembles the kernel trick in the original GE.

KME has been applied successfully in two-sample testing (Gretton et al., 2012; 2009), graphical models (Song et al., 2011), and probabilistic inference (Muandet et al., 2016; Hsu & Ramos, 2019). Recently, learning from distributions in general and KME in particular has drawn a lot of attention in the machine learning community. In (Muandet et al., 2012), KME was used to extend Support Vector Machine (SVM) by making it suitable for learning from distributions. In (Kim & Park, 2018), it was used to develop a kernelization of the classical imitation algorithm proposed in (Abbeel & Ng, 2004). In (Muandet & Schölkopf, 2013), an anomaly detection technique based on Support Measure Machine, which uses KME, was proposed. In (Muandet et al., 2013), a kernel-based domain generalization method was proposed. In general, KME can be seen as a kernelization technique defined on a distribution space. In the next section, we propose using KME for learning non-linear data embeddings under uncertainty.

## 4 Non-linear Graph Embedding with Data Uncertainty

In this section, we present our proposed Non-linear Graph Embedding with Data Uncertainty (NGEU), a framework for non-linear DR designed as an extension of GE to find low-dimensional manifolds in the presence of data uncertainties. In the NGEU learning paradigm, we assume that the uncertainty of each data point is modeled with a probability distribution. Formally, let $\{\mathbb{P}_i\}_{i=1}^N$ be our input data. Inspired by the GE formulation, we assume that the training data form a vertex set $\mathcal{V} = \{v_i\}_{i=1}^N$, where each vertex $v_i$ corresponds to a sample $\mathbb{P}_i$. Based on the pair-wise connections between the vertices, we can now define both the intrinsic and the penalty graphs, as illustrated in Figure 2. The main goal of the non-linear DR approaches is to obtain the mappings of the vertices $\mathcal{V}$ that best 'conserve' the relationships between the vertex pairs. However, as our input space consists of probability distributions, using the kernel trick to learn the non-linear projections directly is not possible. To overcome this limitation, we propose to rely on the KME, which extends the kernel mapping to the distribution space.

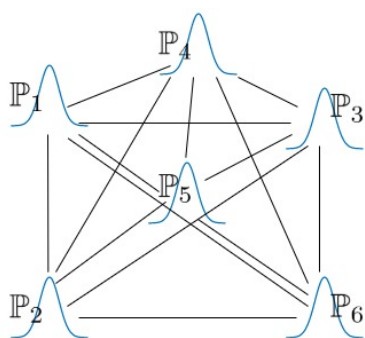

Figure 2: An illustration of the adjacency (or penalty) graph of a dataset composed of six distribution samples $\mathbb{P}_i$. Each distribution (data point) represents a vertex in the graph and is connected to the other vertices via the edges of the adjacency (or penalty) graph.

### 4.1 Formulation of NGEU Objective

Using KME, we can formulate the general objective of NGEU. With each input data distribution $\mathbb{P}_i$, we also associate its class label $c_i$ and its feature mean representation $\mu(\mathbb{P}_i)$ in the RKHS space $\mathcal{H}$ calculated using (6). The graph preserving criterion equation 1 now becomes equivalent to

$$
\begin{aligned}
f^* &= \underset{f}{\arg\min} \ \sum_{i \neq j} (\mathbb{E}_{\mathbb{P}_i}[f] - \mathbb{E}_{\mathbb{P}_j}[f])^2 \mathbf{W}_{ij} \\
&= \underset{f}{\arg\min} \ \sum_{i \neq j} (<\mu(\mathbb{P}_i), f>_{\mathcal{H}} - <\mu(\mathbb{P}_j), f>_{\mathcal{H}})^2 \mathbf{W}_{ij}, \\
&\text{s.t.} \quad U = m,
\end{aligned}
\tag{9}
$$

where $U = \mathbb{E}_{\mathbb{P}}[f]^T \mathbf{B} \mathbb{E}_{\mathbb{P}}[f]$, $\mathbb{E}_{\mathbb{P}}[f] = [\mathbb{E}_{\mathbb{P}_1}[f], \ldots, \mathbb{E}_{\mathbb{P}_N}[f]]^T$, and $\mathbf{B}$ expresses a constraint or a penalty graph as in equation 1.

**Theorem 1.** *(Muandet et al., 2012) Given training examples $(\mathbb{P}_k, c_k) \in \mathfrak{B}_\chi \times \mathbb{R}, k = 1, .., N$, any function $f$ minimizing the function defined in equation 9 admits a representation of the form $f = \sum_k \alpha_k \mu(\mathbb{P}_k)$ for some $\alpha_k \in \mathbb{R}, k = 1, .., N$.*

Theorem 1 corresponds to the Representer theorem for our graph preserving criterion. It indicates how each input uncertainty, i.e., $\mathbb{P}_i$, contributes to the global solution of equation 9. Furthermore, it guarantees that there is always a solution in the linear span of the data points and simplifies the problem search space to a finite dimensional subspace of the original function space which is often infinite dimensional. Thus, our criterion becomes computationally tractable.

**Theorem 2.** *The graph preserving criterion of NGEU is equivalent to*

$$
\begin{aligned}
\boldsymbol{\alpha}^* &= \arg\min_{\boldsymbol{\alpha}} \boldsymbol{\alpha}^T \mathbf{K} \mathbf{L} \mathbf{K}^T \boldsymbol{\alpha}, \\
s.t. \quad &\boldsymbol{\alpha}^T \mathbf{K} \mathbf{L}^{\mathbf{P}} \mathbf{K}^T \boldsymbol{\alpha} = m \quad or \quad \boldsymbol{\alpha}^T \mathbf{K} \boldsymbol{\alpha} = m,
\end{aligned}
\tag{10}
$$

*where $\mathbf{K}$, $\mathbf{L}$, $\mathbf{L^P}$ are the kernel matrix, the Laplacian of the intrinsic graph, and the Laplacian of the penalty graph, respectively.*

*Proof.* Using Theorem 1, the mapping $f$ can be obtained as a linear combination of data in $\mathcal{H}$, i.e., $f = \sum_i \alpha_i \mu(\mathbb{P}_i)$. By defining $\boldsymbol{\alpha} = [\alpha_1, .., \alpha_N]^T$, $\mathbf{K}$ as the kernel matrix associated with $\mu(\mathbb{P}_i)$, i.e. $\mathbf{K}_{ij} =$

$\mathcal{K}(\mathbb{P}_i, \mathbb{P}_j) = <\mu(\mathbb{P}_i), \mu(\mathbb{P}_j)>_{\mathcal{H}}$, and $\mathbf{K}_i$ its $i^{th}$ column, the feature representation $\mathbb{E}_{\mathbb{P}_i}[f]$ can be rewritten as

$$\mathbb{E}_{\mathbb{P}_i}[f] = <\mu(\mathbb{P}_i), f>_{\mathcal{H}} = <\mu(\mathbb{P}_i), \sum_k \alpha_k \mu(\mathbb{P}_k)>_{\mathcal{H}}$$

$$= \sum_k \alpha_k <\mu(\mathbb{P}_i), \mu(\mathbb{P}_k)>_{\mathcal{H}} = \sum_k \alpha_k \mathcal{K}(\mathbb{P}_i, \mathbb{P}_k) = \boldsymbol{\alpha}^T \mathbf{K}_i. \quad (11)$$

$\square$

Since the optimization problem of the proposed framework takes the form in (10), it has one global optimal solution. In fact, the solution of the optimization problem (10) can be obtained by solving the corresponding generalized eigenvalue decomposition problem (5). However, it should be noted here that the solution to kernel GE is different than the solution of NGEU, because the elements of the kernel matrix $\mathbf{K}$ are computed based on the probability distributions, whereas in GE they are obtained based on the discrete input data. Interestingly, the feature map $\mu(\mathbb{P})$ is linear in $\mathbb{P}$, whereas in the standard kernel variant GE, $\phi(\mathbf{x})$ is usually non-linear in $\mathbf{x}$.

It should also be noted that the NGEU framework relies on KME and, as a result, each input distribution $\mathbb{P}$ is mapped to a point, $\mu(\mathbb{P}) \in \mathcal{H}$, whereas in (Laakom et al., 2022) each input distribution is mapped to a Gaussian distribution. This yields a different embedding even for NGEU with a linear kernel. In fact, using equation 8, the elements of the linear kernel matrix $\mathbf{K}$ can be computed as $\mathbf{K}_{ij} = \mathcal{K}(\mathbb{P}_i, \mathbb{P}_j) = \boldsymbol{\mu}_i^T \boldsymbol{\mu}_j + \delta_{ij} \text{Tr}(\text{Var}(\mathbb{P}_i))$ for any arbitrary distributions $\mathbb{P}_i$ and $\mathbb{P}_j$, where $\delta_{ij}$ is the Kronecker delta function equal to one if i is equal to j and zero otherwise. Compared to GE, we note that by incorporating uncertainty the graph preserving criterion of NGEU with the linear kernel yields a diagonal regularisation of the kernel matrix of GE. In the extreme case, i.e., using a Dirac distributions for each sample, NGEU becomes equivalent to the original GE.

**Theorem 3.** *Given training samples $(\mathbb{P}_i, c_i) \in \mathfrak{B}_\chi \times \mathbb{R}, i = 1, .., N$, where $\mathbb{P}_i$ can be any arbitrary distribution other than Dirac distribution, modeling the uncertainty of the $i^{th}$ sample, the matrices involved in the objective function equation 10 with the linear kernel, i.e., matrices $\mathbf{KLK}^T$ and $\mathbf{KL^pK}^T$, have equal or higher rank than those of the GE (Equation (4)).*

The proof is available in the Appendix. The minimization in (4) of GE is an *ill-posed* problem in the case of the linear kernel matrix, as it is only positive semi-definite and not positive definite. Theorem 3 implies that for NGEU, the data uncertainty acts as a regularizer and increases the rank of the matrices. Thus, the use of uncertainty under the proposed framework provides a data-driven mechanism to regularize the scatter matrices of the graphs and increase their ranks. This yields more relevant projection dimensions compared to the original DR methods.

The proposed NGEU framework can also be interpreted as a kernelization of (Laakom et al., 2022) using the kernel mean embedding. This makes NGEU suitable to capture meaningful manifolds of non-linearly distributed data in the presence of uncertainty. As explained above, one key limitation of (Laakom et al., 2022) is the assumption that the input data uncertainty is modeled by a Gaussian distribution. If this constraint is violated, the graph preserving criterion cannot be expressed in terms of the original input data. In NGEU, this constraint is removed and the graph preserving criterion can be characterized for any type of distribution $\mathbb{P}$ as long as $\mathcal{K}(\mathbb{P}_i, \mathbb{P}_j)$ can be computed. As a result, NGEU provides more flexible modeling of the input uncertainty compared to (Laakom et al., 2022) for the different problems raised in the literature.

Note that in this paper, we do not advocate for any particular DR approach. In fact, the proposed framework NGEU, as an extension of general GE, can be used to extend several traditional non-linear DR methods, such as KPCA and KMFA, to incorporate data uncertainty.

## 4.2 Learning Guarantees for NGEU

In this section, we analyze how the uncertainty information theoretically affects the generalization ability of the methods and study the generalization performance of the proposed framework NGEU. Several techniques

have been proposed to study the generalization of different approaches (Mohri et al., 2015; Cortes et al., 2013; Mosci et al., 2007). In this work, we rely on the Rademacher complexity, which is defined as follows:

**Definition 1.** *(Bartlett & Mendelson, 2002) Let $S$ be a dataset formed by $N$ samples $\{\boldsymbol{x}_i\}_{i=1}^N$ from a distribution $\mathcal{Q}$. Let $\mathcal{F}$ be the hypothesis class, i.e., $\mathcal{F}: \mathcal{X} \to \mathbb{R}$. Then, the empirical Rademacher complexity $\mathcal{R}_S(\mathcal{F})$ of $\mathcal{F}$ is defined as follows*

$$\mathcal{R}_S(\mathcal{F}) = \mathbb{E}_\sigma\left[\sup_{g\in\mathcal{F}} \frac{1}{N}\sum_{i=1}^N \sigma_i g(\boldsymbol{x}_i)\right],$$

*where $\sigma = \{\sigma_1, \cdots, \sigma_N\}$ are independent uniform random variables in $\{-1, 1\}$.*

The Rademacher complexity is a data-dependent learning theoretic notion that is used to measure the richness of a hypothesis class. It is used as a proxy of generalization (Mohri et al., 2015; Gottlieb et al., 2016; Von Luxburg & Schölkopf, 2011), i.e., it quantifies the difference between the train and test error. In this work, we want to to understand the effect of uncertainty on the generalization ability of the models. To this end, we start by comparing the Rademacher complexity of the solutions of NGEU and GE in the presence of noise in the dataset:

**Theorem 4.** *Given training samples $\mathbb{S} = \{(\mathbb{P}_k, c_k) \in \mathfrak{B}_\chi \times \mathbb{R}, k = 1, .., N\}$, the associated hypothesis set of the solution of NGEU, i.e., equation 10, has the form $\mathcal{F}^{NGEU} = \{\mathbb{P} \to < \boldsymbol{w}, \mu(\mathbb{P}) >\}$. The corresponding Rademacher complexity of NGEU is upper-bounded as follows:*

$$\mathcal{R}_\mathbb{S}(\mathcal{F}^{NGEU}) \le \mathbb{E}_{S\sim\mathbb{S}}\big[\mathcal{R}_S(\mathcal{F}^{GE})\big], \tag{12}$$

*where $S \sim \mathbb{S}$ refers to $S = (\boldsymbol{x}_1, \cdots, \boldsymbol{x}_N) \sim (\mathbb{P}_1, \cdots, \mathbb{P}_N)$, $\mathcal{F}^{GE} = \{\boldsymbol{x} \to < \boldsymbol{w}, \phi(x) >\}$ is the associated hypothesis set of the solution of GE, i.e., (4), and $\mathcal{R}_S(\mathcal{F}^{GE})$ is the Rademacher complexity of $\mathcal{F}^{GE}$.*

The proof is available in the Appendix. Theorem 4 states that in the presence of noise the Rademacher complexity of NGEU is less than the expectation over the sample distributions of the Rademacher of the hypothesis class of corresponding GE. So if the noise is accurately estimated, taking it into account in the solution as in NGEU reduces the Rademacher complexity of the hypothesis class of the methods. Moreover, as lower Rademacher complexity is associated with lower generalization error (Gottlieb et al., 2016; Mohri et al., 2015), Theorem 4 shows that incorporating the uncertainty information yields better generalization performance compared to GE. We note also that in the extreme case, when the sample distributions are Dirac distributions around the original measurements, i.e., $\mathbb{P}_i = Dirac(\mathbf{x}_i)$ for every $i \in 1..N$, both terms of the inequality in Theorem 4 become equal and our framework becomes equivalent to GE.

The Rademacher complexities of classical kernel-based hypotheses defined over a discrete input space are typically upper-bounded using the trace of the associated kernel and the norm of the projection vector (Cortes et al., 2013; Gottlieb et al., 2016). Theorem 5 extends this classic bound to the distribution space and shows that the bound remains valid for the distribution input space of NGEU. We show that the empirical Rademacher complexity of NGEU with bounded norm, i.e., with the form $\mathcal{F} = \{\mathbb{P} \to < \mathbf{w}, \mu(\mathbb{P}) >, ||\mathbf{w}|| < A\}$ can be bounded using the trace of the kernel matrix.

**Theorem 5.** *Given training samples $\mathbb{S} = \{(\mathbb{P}_k, c_k) \in \mathfrak{B}_\chi \times \mathbb{R}, k = 1, .., N\}$ and a kernel $\mathcal{K} : \mathbb{P} \times \mathbb{P} \hookrightarrow \mathbb{R}$ associated with the feature map $\mu(\mathbb{P})$, the Rademacher complexity of the bounded hypothesis set of NGEU, i.e., $\mathcal{F} = \{\mathbb{P} \to < \boldsymbol{w}, \mu(\mathbb{P}) >, ||\boldsymbol{w}|| < A\}$, can be upper-bounded as follows:*

$$\mathcal{R}_\mathbb{S}(\mathcal{F}) \le A\frac{\sqrt{Tr(\mathbf{K})}}{N}, \tag{13}$$

*where $Tr(\cdot)$ is the trace operator and $\mathbf{K}$ is the kernel matrix in equation 10.*

The proof is available in the Appendix. Theorem 5 bounds the the Rademacher complexity of the hypothesis set of NGEU using the probabilities of self-similarities, i.e., $Tr(\mathbf{K}) = \sum_i \mathcal{K}(\mu(\mathbb{P}_i), \mu(\mathbb{P}_i))$. Moreover, it shows that, under practical assumptions, providing more training data, larger $N$, yields lower complexity and, hence, better generalization performance for NGEU. We also note that the upper-bound is fully data-dependent and can be computed directly using the trace of the kernel matrix.

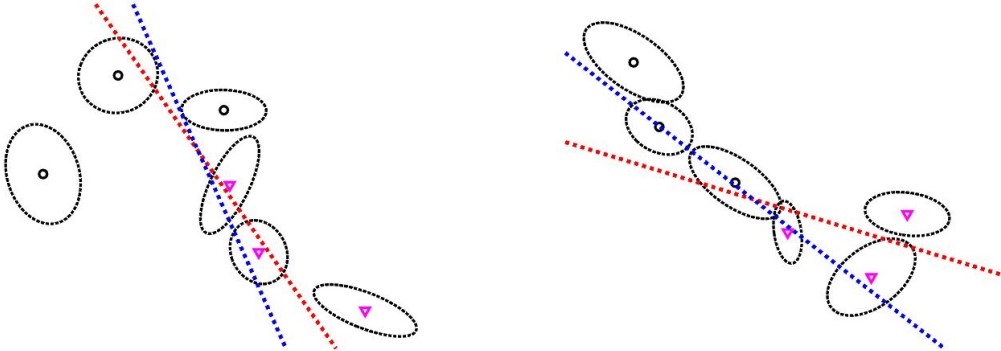

Figure 3: Projection directions obtained by GE (red) and our NGEU (blue) using the graphs of LDA with linear kernel (left) and the graphs of MFA (right) on a randomly generated synthetic 2-D data from two classes (circle and triangle). The data contains three samples per class generated with respective uncertainty shown as ellipses around each data point.

To better understand the effect of incorporating the uncertainty information, we derive the upper-bound in Theorem 5 for the special case of NGEU with the linear kernel in Theorem 6.

**Theorem 6.** *Let* $S = \{(\boldsymbol{x}_k, c_k) \in \mathbb{R}^D \times \mathbb{R}, k = 1, .., N\}$ *be a training dataset and let* $\mathbb{S} = \{(\mathbb{P}_k, c_k) \in \mathfrak{B}_\chi \times \mathbb{R}, k = 1, .., N\}$ *be the dataset modeling the uncertainty of* $S$ *so that each sample* $\boldsymbol{x}_k$ *is modeled with a distribution* $\mathbb{P}_k$ *with a mean* $\mathbb{E}[\mathbb{P}_k] = \boldsymbol{x}_k$ *and a finite variance* $\Sigma_k$ *modelling the uncertainty of the sample. The Rademacher complexity of NGEU with linear kernel is bounded as follows:*

$$\mathcal{R}_{\mathbb{S}}(\mathcal{F}^{NGEU}) \leq \frac{A}{N}\sqrt{Tr(\mathbf{K^{GE}})} + \frac{A}{N}\sqrt{\sum_i Tr(\Sigma_i)} = \mathcal{R}_S(\mathcal{F}^{GE}) + \frac{A}{N}\sqrt{\sum_i Tr(\Sigma_i)} \qquad (14)$$

*where* $\mathcal{R}_S(\mathcal{F}^{GE})$ *is the Rademacher complexity of the original GE with training samples* $S$.

*Proof.* By using equation 8, we have $K(\mathbb{P}_i, \mathbb{P}_j) = \boldsymbol{\mu}_i^T \boldsymbol{\mu}_j + \delta_{ij} \text{Tr}(\boldsymbol{\Sigma}_i)$ and the result is straightforward from Theorem 5. □

The upper-bound found in Theorem 6 is composed of two terms. The first term is the typical bound of the kernel-based hypothesis and it does not depend on the uncertainty information, whereas the second term depends on the samples' variance. The more certain we are about the data, the smaller the second term is and, thus, the bound is tighter. Intuitively, in the case of highly uncertain data, due to measurement errors for example, it becomes harder to learn robust and meaningful projections and, thus, to generalize well. On the other hand, in the extreme case where each sample is expressed by a Dirac distribution, i.e., there is no uncertainty, the second term becomes zero and the bound is tighter.

## 5 Experiments and Analysis

### 5.1 Toy Example

We start our empirical analysis with a 2-D synthetic data binary problem to provide insights regarding how the proposed framework works. We experiment with the variants of LDA and MFA for the standard GE and their counterparts derived using our framework, i.e., NGEU, using the linear kernel. For illustration purposes, we generate random Gaussian uncertainty for each sample. The input data, the uncertainty and the methods' outputs are presented in Figure 3.

As illustrated in the figure, incorporating uncertainty indeed shifts the projection directions and the outputs generated by GE is different than those of our framework for both MFA and LDA. The standard GE does

not consider the input noise information and, thus, it risks generating non-robust embedding of the data. NGEU, on the other hand, leverages the additional information on the uncertainty, presented in the form of distribution, to generate a more robust embedding of the data.

## 5.2 Image Classification

In this section, we evaluate the performance of the proposed framework on several image classification tasks. We consider the three different DR approaches LDA, PCA, and MFA using the original Graph Embedding (GE) (Yan et al., 2007), Graph Embedding with Data Uncertainty (GEU) (Laakom et al., 2022), and our proposed framework Non-linear Graph Embedding with Data Uncertainty (NGEU). We denote by LDA-GE, LDA-GEU, and LDA-NGEU the LDA variants formulated using GE, GEU, and our framework, respectively. We denote the variants of PCA and MFA in a similar manner. For the uncertainty estimation, we use the approach proposed in (Laakom et al., 2022), i.e., each input sample is modeled using a normal distribution with a mean equal to the original data point and a variance based on its distance to the nearest other data point. After the DR step, we apply a K-Nearest Neighbor classifier (k-NN) (Cover & Hart, 1967) with $k = 5$. For comparative purposes, we also add the performance of a Neural Network with 1 intermediate layer of size 128 with ReLU activation (NN-1), Neural Network with 2 intermediate layer of size 128 with Relu activation (NN-2),SVM (Cortes & Vapnik, 1995), Decision Tree classifier (Myles et al., 2004), Nearest Neighbor on the original data, and K-Nearest Neighbor applied after DR using Trimmed Robust Principal Component Analysis (TRPCA) (Podosinnikova et al., 2014), Robust Sparse Linear Discriminant Analysis (RSLDA) (Wen et al., 2018), and Fast Subclass Discriminant Analysis (fastSDA) (Chumachenko et al., 2020; Chumachenko et al., 2021). For the neural network models, we train for 100 epoch using SGD and weight decay of $1e^{-4}$.

To evaluate the performance of these methods, we use four standard classification datasets: COIL20, MNIST, USPS, and ORL. COIL20 dataset is an image dataset containing images of 20 objects[2]. For each object, there are 72 images in total and the size of each image is $32 \times 32$ pixels, each represented by 8 bits. Thus, each image is represented by a 1024-dimensional vector. In our experiments, we train all the methods on the first 55 images of each object (1100 training images in total), validate on the next 5 samples of each object (100 in total), and the rest is used for testing (240 in total). The MNIST dataset is a handwritten digit dataset composed of 10 classes. MNIST images are $28 \times 28$ pixels, which results in 784-dimensional vectors. We use a subset of this dataset composed of 2000 training images and 2000 test images (500 for validation and 1500 for testing). The USPS handwritten digit dataset is described in (Hull, 1994). A popular subset[3] contains 9298 $16 \times 16$ images in total. It is split into 7291 training images and 2007 test images (Cai et al., 2011; 2010). In our experiments, we train all the methods on the first 6000 images of the training set, validate on the last 1000 samples of the training set, and test on the 2007 test images. The ORL dataset (Samaria & Harter, 1994) contains images from of 40 distinct classes taken under different conditions. For each class, we have only 10 samples, which makes it a challenging dataset. In our experiment, we use 5 samples per class for training, 2 per class for validation, and 3 samples per class for testing.

In our experiments, we select the best hyper-parameter values on the validation set for each approach and use these values in the testing phase. For PCA and MFA variants, the dimension $d$ of the subspace is selected from $\{1, 2, 4, 8, 16, 32\}$ for the four datasets. For LDA, the maximum projection directions for LDA-GE is constrained by the total number of classes. Thus, for a fair comparisons of LDA variants, $d$ is selected from $\{1, 2, 4, 8, 16\}$ for COIL20 and ORL and from $\{1, 2, 4, 6, 8\}$ for USPS and MNIST. For the uncertainty estimation, in both our framework NGEU and GEU, there is one hyper-parameter $\lambda$ characterizing the width of the distribution. This value is selected from $\{0.001, 0.1, 0.2, 0.4, 0.8, 1, 2\}$. For the kernel based approaches, we experiment with three kernels, linear, RBF, and second degree polynomial, and we select the one with the highest validation accuracy. For the RBF kernel, the hyper-parameter $\sigma$ is selected from $\{0.1, 1, 4, 16, 32, 64, 100\}$.

In Table 1, we report the results of different approaches on the four datasets. For supervised DR case, i.e., LDA and MFA, we note that in general frameworks incorporating uncertainty in the learning process, i.e.,

---

[2]cs.columbia.edu/CAVE/software/softlib/coil-20.php
[3]csie.ntu.edu.tw/ cjlin/libsvmtools/datasets/multiclass.html

GEU and our framework NGEU, consistently outperform the counterpart standard variants of GE. For the unsupervised method, i.e., PCA, GE achieves the best performance over the four datasets. We also note that our proposed framework consistently achieves higher accuracy than GEU with the exception of PCA on COIL20 and MNIST. In fact, for the MFA and the LDA variants, NGEU achieves $> 4\%$ accuracy boost compared to GEU across all datasets. While also GEU leverages the uncertainty information, it is able to capture only the linear projections as opposed to our framework, which can capture non-linear transformations of the data.

Table 1: Accuracy of the different approaches on the datasets

|  | COIL20 | MNIST | USPS | ORL |
|---|---|---|---|---|
| NN-1 | 93.75% | 84.73% | 91.28% | 81.67% |
| NN-2 | 91.25% | 84.46% | 91.45% | 78.33% |
| SVM | 93.75% | 87.87% | 94.17% | 58.33% |
| Decision Tree | 72.08% | 68.27% | 83.47% | 41.67% |
| k-NN | 88.75% | 87.27% | 94.57% | 70.00% |
| TRPCA+k-NN | 94.58 % | 89.13% | 94.37% | 71.66% |
| RSLDA+k-NN | 80.00% | 83.87% | 93.68% | 80.00% |
| fastSDA+k-NN | 95.42% | 83.67% | 91.09% | 68.33% |
| KMFA-GE+k-NN | 84.17% | 79.27% | 89.99% | 78.33% |
| MFA-GEU+k-NN | 84.16% | 84.73% | 86.25% | 71.67% |
| KMFA-NGEU+k-NN | 92.92% | 89.00% | 90.04% | 78.33% |
| KDA-GE+k-NN | 88.33% | 81.60% | 88.40% | 76.67% |
| LDA-GEU+k-NN | 95.42% | 84.73% | 90.34% | 76.67% |
| KDA-NGEU+k-NN | 98.75% | 91.07% | 94.42% | 81.67% |
| KPCA-GE+k-NN | 92.50% | 90.40% | 94.17% | 70.00% |
| PCA-GEU+k-NN | 92.50% | 89.13% | 89.54% | 65.00% |
| KPCA-NGEU+k-NN | 92.08% | 88.53% | 91.43% | 68.33% |

We note that by incorporating uncertainty in the GE framework, we obtain a significant boost in performance for KDA and KMFA across all the datasets: $> 9\%$ boost in accuracy for KDA and KMFA on the MNIST dataset and $> 8\%$ on the COIL20 dataset. However, KPCA fails to gain improvement over the baseline, i.e., GE. This might be because KPCA does not encode the discriminative information. Moreover, we note that using a Gaussian modeling of the uncertainty might not be optimal for the image classification task. Nonetheless, it yields more robust variants of the classical DR methods, KDA and KMFA. Using a more descriptive distribution for image uncertainty should lead to further improvement.

### 5.3 Noisy Data

In this section, we test our framework on noisy data. To this end, we use AWGN-MNIST (Basu et al., 2017): a noisy variant of MNIST. The dataset is publicly available[4] and has the same structure as the original MNIST. It is created using additive white Gaussian noise with signal-to-noise ratio of 9.5. In this paper, we use the subset formed by the first 2000 training samples of this dataset. We use this subset to evaluate our methods using a three-fold cross validation. We report the mean and the standard deviation of the accuracy on the three folds. In addition to the uncertainty estimation protocol of (Laakom et al., 2022), here we also experiment with a basic uncertainty schema: Every sample $\mathbf{x}$ is modeled by a Gaussian uncertainty $\mathbb{P}_G = \mathcal{N}(\boldsymbol{\mu}, \boldsymbol{\Sigma})$ with a mean $\boldsymbol{\mu} = \mathbf{x}$ and a constant variance along all the feature directions: $\boldsymbol{\Sigma} = \alpha \mathbf{I}$, where $\alpha$ is a constant and $\mathbf{I}$ is the identity matrix. For the hyper-parameter selection, we use the same protocol as the previous section with the standard MNIST. The hyper-parameter $\alpha$ of the basic uncertainty is selected from $\{0.001, 0.005, 0.01\}$. We denote the NGEU approaches using this noise estimate as X-NGEU-N+k-NN.

In Table 2, we report the mean and the standard deviation of the accuracy achieved by the different methods on the AWGN-MNIST dataset. For KLDA and KMFA, we note that by leveraging the uncertainty infor-

---

[4]https://csc.lsu.edu/~saikat/n-mnist/

Table 2: Mean and standard deviation of the accuracy of the different approaches on AWGN-MNIST

|  | AWGN-MNIST |
| --- | --- |
| SVM | $82.75 \pm 1.29$ % |
| Decision Tree | $39.70 \pm 1.88$ % |
| k-NN | $84.15 \pm 1.34$ % |
| TRPCA + k-NN | $88.05 \pm 0.39$ % |
| RSLDA + k-NN | $52.85 \pm 2.01$ % |
| fastSDA + k-NN | $84.80 \pm 1.18$ % |
| KMFA-GE+k-NN | $65.55 \pm 5.10$ % |
| MFA-GEU+k-NN | $48.40 \pm 7.40$ % |
| KMFA-NGEU+k-NN | $75.20 \pm 1.52$ % |
| KMFA-NGEU-N+k-NN | $79.40 \pm 0.70$ % |
| KDA-GE+k-NN | $64.95 \pm 1.04$ % |
| LDA-GEU+k-NN | $81.75 \pm 3.17$ % |
| KDA-NGEU +k-NN | $81.45 \pm 1.22$ % |
| KDA-NGEU-N+k-NN | $82.55 \pm 1.66$ % |
| KPCA-GE+k-NN | $88.25 \pm 1.01$ % |
| PCA-GEU+k-NN | $88.25 \pm 0.89$ % |
| KPCA-NGEU+k-NN | $87.45 \pm 0.88$ % |
| KPCA-NGEU-N+k-NN | $87.75 \pm 0.43$ % |

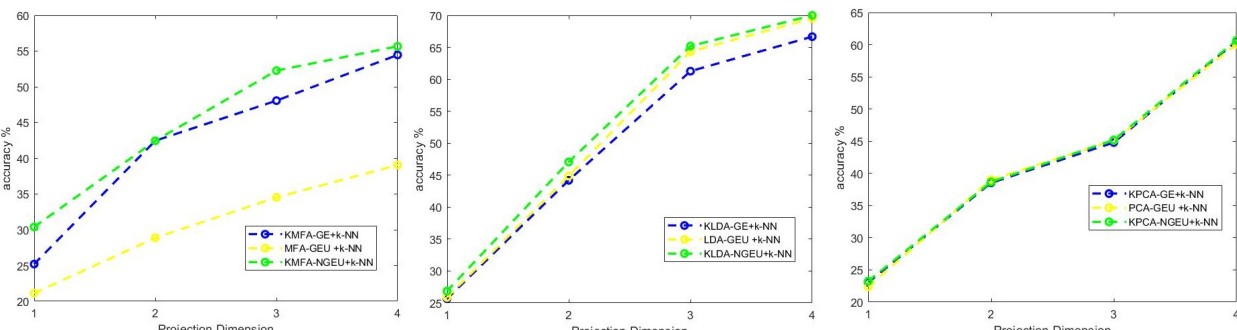

Figure 4: Final accuracy as a function of projection dimension for GE, GEU, and NGEU variants of KMFA, KLDA, and KPCA on AWGN-MNIST dataset.

mation our framework boosts the results of the GE. In fact, NGEU yields more than 10% mean accuracy boost for MFA and more than 15% for LDA. GEU is able to learn only linear projections and, thus, it fails if non-linearity is required as in the case of MFA-GEU. We note that, similar to the results with the non-noisy datasets, including uncertainty does not improve the results for unsupervised DR, i.e., KPCA. This can be explained by the absence of discriminant information in KPCA. Using better uncertainty modeling techniques can improve the results in this case. It is also worth mentioning that our framework achieves more robust and consistent performance. This can be seen by the low standard deviation, i.e., lower than 1.7%, for all the variant compared to 5.1% standard deviation for MFA with GE. We also note that here the constant Gaussian noise estimate $\mathbb{P}_G$ yields better results than the protocol adopted from (Laakom et al., 2022). This can be explained by the fact that AWGN-MNIST is constructed using Gaussian noise.

To understand how the projection dimension $d$ affects the performance of the different DR frameworks, i.e., GE, GEU, and NGEU, we report the accuracy of the methods for different dimensions. The results are presented in Figure 4. As it can be seen, the results are consistent with our findings in Table 2, i.e., our approach consistently boosts the performance of the DR methods, especially KMFA.

# 6 Conclusion

In this paper, we introduced a novel nonlinear dimensionality reduction framework that can leverage the available input data uncertainty information presented in the form of probability distributions. The proposed framework, called Nonlinear Graph Embedding with Data Uncertainty (NGEU), has one global solution and can find nonlinear low-dimensional manifolds in the presence of data uncertainties and artifacts. Our framework can incorporate a multitude of distributions, enabling a flexible modeling of the uncertainty. Moreover, we provide learning guarantees for our framework based on the Rademacher complexity of its hypothesis class and we show how uncertainty affects the generalization of the approaches. Experimental results demonstrate the effectiveness of the proposed framework and show that incorporating uncertainty yields $> 6\%$ accuracy boost for LDA and MFA across multiple datasets.

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
