# OpenReview forum: "Non-Linear Spectral Dimensionality Reduction Under Uncertainty"
_TMLR — Rejected by TMLR_

### Review · Reviewer_ZYfR · 2024-12-28

**Summary Of Contributions:**

This paper introduces a new framework, Non-linear Graph Embedding with Data Uncertainty (NGEU), which extends traditional dimensionality reduction (DR) techniques by incorporating uncertainty information.
The NGEU framework models data points as probability distributions and embed (i.e., GE) them into a hilbert space, enabling a formulation as an optimization problem in this space (as stated in Theorem 1); by solving this optimization problem they obtained a robust dimensionality reduction in the presence of (gassian) noise and measurement uncertainty in the data points.

NGEU framework generalizes the Graph Embedding (GE) approach to handle uncertainty, extending classical methods like KPCA KDA and KMFA. Experimental results demonstrate significant performance improvements across multiple datasets, particularly for supervised methods like LDA and MFA, where incorporating uncertainty leads to more than a 5% accuracy boost.

**Audience:**

Yes

**Broader Impact Concerns:**

This paper addresses an important problem in (nonlinear) dimensionality reduction by introducing a framework that explicitly accounts for (Gaussian assumtion) uncertainty in data. Its contributions are well-supported by theoretical guarantees and validated by experimental results on real-world datasets. However, the lack of comparisons with popular methods like t-SNE and UMAP and limited exploration of scalability and kernel impacts narrow its audience.

**Claims And Evidence:**

Yes

**Requested Changes:**

(1)Please add comparisons with t-SNE and UMAP. These methods are widely adopted for non-linear DR and perform well on datasets like MNIST, CIFAR-10 and USPS. Including them in experimental evaluations would provide a more comprehensive benchmark and make the paper appealing to a broader audience. Actually, I think KMFA is not a usual competitor in nonlinear DR.

(2)  How scalable this method is for high-dimensional datasets? For example, testing NGEU on datasets with thousands of features and discussing how computational or memory constraints might arise in these cases. And, moreover, how the computational time scales with the sample size and dimensionality, should be discussed carefully (typically there will be a big O complexity analysis, but considering the emphasis of TMLR, this would be optional but helpful). This will clarify the practical upper limits of the method.

(3) Will this current method work well under alternative distributions for modeling uncertainty? The experiments focus on Gaussian noise, but including other types of noise or uncertainty, such as uniform or multimodal distributions, I would suggest to demonstrate the flexibility and robustness of the framework.

(4) Please provide more clarity on the effect of projection dimensions. While some results are shown in Figure 4, a more detailed discussion of how varying dimensions affect performance across datasets and DR methods would be useful. This analysis could include insights into the trade-offs between dimensionality and accuracy.

(5) The authors could highlight specific computational benefits of the closed-form solution. For instance, include a complexity analysis that quantifies how the closed-form approach improves efficiency compared to iterative methods. This would strengthen the claims of computational scalability. There are recent research on closed-form non-linear DR methods (e.g., https://www.jmlr.org/papers/v25/23-0547.html; the same paper also discussed consistency in dimension reduction context) with detailed analysis using subsequent tasks, it would be enlightening to see how NGEU compared to more specific closed-form nonlinear DR approaches.

Points (1), (4) and (5) are critical; other points can greatly strengthen the paper.

**Strengths And Weaknesses:**

Strengths
(1)The paper offers a novel way to incorporate uncertainty directly into dimensionality reduction, which is a key challenge in real-world noisy data. The flexibility to handle various types of distributions extends its applicability beyond existing methods like GEU.
(2) The closed-form solution for NGEU indeed ensures computational efficiency, a crucial factor when working with high-dimensional datasets and probabilistic modeling. But also see my point (5) of requested changes below.

Weaknesses
(1) The paper does not compare the performance of NGEU with widely used modern dimensionality reduction techniques like t-SNE and UMAP, which are highly relevant for non-linear data embeddings. This omission weakens the appeal to practitioners and researchers familiar with these methods (like when shall we use NEGU instead of more traditional methods).

(2) The use of Rademacher complexity to analyze generalization is a theoretical contribution. It ties the framework to well-established principles in learning theory. But I am not entirely sure how relevant it is in dimension reduction: I agreee that with Gaussian distributional assumption we may discover the true solution to equation (4), but that seems to be more of an optimization result instead.

---

### Review · Reviewer_5sym · 2024-12-30

**Summary Of Contributions:**

This paper proposed Non-Linear Graph Embeddings under Data Uncertainty (NGEU) as a mean for dimension reduction while incorporating the sampling distribution information. The method combines non-linear embeddings and data uncertainty with kernel mean embedding and optimize over a graph matching objective to preserve graph information. This paper also provide generalization guarantees of NGEU in the form of Rademacher complexity of such transformations, which could be used to bound downstream learning tasks. Finally, the paper provides simulations on synthetic and real-world datasets to show the potential gain of NGEU in learning tasks.

**Audience:**

Yes

**Claims And Evidence:**

No

**Requested Changes:**

See Weakness.

**Strengths And Weaknesses:**

Strength:
1. The structure of the paper is well organized and easy to follow.
2. Invoking non-linearity in embedding transformation with kernel trick is standard in distributional learning and makes sense.
3. Although the proof of generalization is not included in the submission, the order (with respect to n) looks correct.

Weakness:
1. Graph-based Kernel Mean Embedding is not a new idea in distributional learning, however, the author did not discuss literature on this topic, which is closely related to what was proposed in this paper. The authors should also pinpoint the novelty of this work compared to existing graph-based kernel mean embeddings when doing so.
2. My main concern is the validity of the premise of the work. The target problem of this paper is not much different from standard kernel mean embedding techniques in distributional learning, except that the authors name the study on the distribution as uncertainty. However, in distributional learning, the distribution information is acquired through an empirical estimation based on multiple data points from a distribution, while in this paper, it seems the distribution information is given. I cannot imagine a practical case that one has access to the distribution of each individual data points. The authors should highlight the validity of the target problem, give some practical motivations and also explain how it differs from distributional learning.
3. In kernel mean embedding literature, the choice of kernel determines the information preserved from a distribution after the transformation. In fact, the distribution information will only be fully preserved if the kernel function is characteristic. In another word, a bad choice of kernel might completely ignore the uncertainty of the data distribution, which makes the non-linearity contradict with data uncertainty. However, I did not see any discussion regarding this matter.
4. The Rademacher complexity results' usefulness are questionable, as it appears to be standard results for kernel functions and became meaningless for shift-invariant kernel like RBF kernels (i.e., Tr(K) = N). The authors should provide the proof for the reviewers to check for correctness and usefulness.
5. KDA-NGEU appeared multiple times in the results without being formally introduced, if this method is really KDA based, doesn't that invalidate the comparison with LDA based method? In general, the explanation on experimental details and results discussion in this paper is not satisfactory.
6. A minor issue regarding the $\sigma$ parameter for RBF kernel in the simulations, the validation values are in general too large for MNIST based dataset in my experience, can the author show the optimal $\sigma$ under cross-validation?


Overall:
The paper shows a lack of rigor throughout the writing and results, which in my opinion, is important for TMLR standard. I cannot recommend acceptance at this point.

---

### Review · Reviewer_K2kg · 2025-01-07

**Summary Of Contributions:**

This paper introduces a novel framework called Non-linear Graph Embedding with Data Uncertainty (NGEU) for dimensionality reduction that incorporates input data uncertainty. The key contributions are:
- A new non-linear dimensionality reduction framework that can leverage input data uncertainty presented as probability distributions.
Extension of the Graph Embedding (GE) framework to handle uncertain inputs using kernel mean embeddings.
- Ability to create uncertainty-aware versions of methods like KPCA, KDA, and KMFA.
- Theoretical analysis providing learning guarantees based on Rademacher complexity, showing how incorporating uncertainty can improve generalization performance.
- A closed-form global solution for the proposed framework.
- Flexibility to work with various types of input distributions, not limited to Gaussian uncertainty.

**Audience:**

Yes

**Broader Impact Concerns:**

I do not have any concerns on this part.

**Claims And Evidence:**

No

**Requested Changes:**

1. Please see the weakness part mentioned before.
2. On page 6, you mention 'Since the optimization problem of the proposed framework takes the form in (10), it has one global optimal solution.' Please add some explanations for this one. (at least you need to give some reference). Furthermore, could you elaborate on the complexity of calculating the global optimum of the problem (10)?
3. On page 8, 'As illustrated in the figure, incorporating uncertainty indeed shifts the projection directions and the outputs generated by GE is different than those of our framework for both MFA and LDA.' I personally think the role of Figure 3 is **unclear**. There is no more insight than just showing the difference. You can consider removing Figure 3 or giving more insights in different ways using toy examples.

**Strengths And Weaknesses:**

Strengths
- Novel approach: Incorporates uncertainty in non-linear dimensionality reduction
- Theoretical foundations: Provides learning guarantees and complexity analysis
- Flexibility: Can work with various types of input distributions
- Empirical validation: Demonstrates improvements on multiple datasets and tasks

Weakness:
- As the paper mentioned, the global optimum guarantee for the problem (9) is the highlight of this paper. As I saw in the proof, the global optimality proof relies on Theorem 2, a corollary of Theorem 1. Can you explain more on this point? Why is the global optimum so beneficial? (e.g. in other works/literature, do they have algorithms for certain optimization problems that can not reach the **global optimum**(e.g. a nonconvex optimization problem with unclear landscape)? ) The transition from (9)-(10) is based on the Representation Theorem to rewrite the optimization problem. From my point of view, it is a little bit **overclaimed** for repeatedly claiming **global optimality**. I am happy to discuss this with the authors in the future discussion period.
- For the complexity proof in this paper, I am not sure whether the author can give some **generalization performance comparison** between algorithms in the paper and other types of algorithms for the same task. As I saw in the paper, the author mentioned there are **no prior works** that possess theoretical guarantees for the task. I wonder whether your framework in the proof can give some insights for analyzing other algorithms in the same field.
- For the numerical experiments, the paper primarily uses Gaussian uncertainty, which may not be optimal for all data types. The authors admit this limitation, stating that "using a Gaussian modeling of the uncertainty might not be optimal for the image classification task." The proposed NGEU framework fails to improve performance for unsupervised dimensionality reduction methods, particularly KPCA. The authors admit that "KPCA fails to gain improvement over the baseline, i.e., GE" across all datasets tested. I wonder whether you can do more experiments on some **modern datasets** to convince readers of the power of KPCA.
- Details: In Theorem 5, you use *A* with no description, could you elaborate this term?

---

### Review · Reviewer_yZfm · 2025-02-02

**Summary Of Contributions:**

This paper addresses a problem with limited existing solutions: non-linear dimensionality reduction. The proposed method extends the Graph Embedding (GE) framework and leverages available uncertainty information in input data. The approach also allows for the adaptation of other dimensionality reduction methods. It provides theoretical learning guarantees and analyzes how incorporating uncertainty affects the generalization ability of the methods, based on Rademacher complexity. The empirical results demonstrate that the proposed approach outperforms state-of-the-art solutions.

**Audience:**

Yes

**Claims And Evidence:**

No

**Requested Changes:**

- In the Introduction "We propose NGEU ... an extension of GE" : Provide the full name along with the acronym.
- Summarize the work of NGEU to better position the contributions.
- Adding an algorithm would help in understanding the approach.
- 4.1 Formulation of NGEU Objective : define $f$; Provide the definition sets of *i* and *j*.
- I think that Theorem 2 is more of a proposition.
- Add more runs for the experiments as well as the uncertainty of the performance metrics.
- Table 1: Bolden the proposed approach (or 'Ours').

**Strengths And Weaknesses:**

Strengths:
- Learning Guarantees for NGEU
- A new dimensionality reduction framework: can be used to extend several methods
- The effectiveness of the proposed framework: yields promising results
- provides more flexible modeling of the input uncertainty

Weaknesses:
- The proposed framework is interesting but appears to be an extension of the work by Laakom et al. (2022), incorporating the kernel trick. Indeed, the Graph Embedding Data Uncertainty approach can be adapted to LDA, PCA, and MFA, just as NGEU extends to kLDA, kPCA, and kMFA through the kernelized GE framework.
- Some sections do not seem very clear and would benefit from a revision to improve the understanding of the approach (cf requested changes).
- The appendices are missing.
- The experiments differ from those in the paper by Laakom et al. (2022).
- The experiments are based on a single run, and the uncertainty around accuracy is missing. This could be addressed using a k-fold approach, for example. Moreover, the results are not significantly better than those of other approaches or the baseline.
- The k-fold approach for noisy data is limited to 3 folds, which may be insufficient to capture the variability and robustness of the proposed method.

---

### Decision · Action_Editor_moRn · 2025-03-21

**Recommendation:** Reject

**Comment:**

This paper proposes Non-Linear Graph Embeddings under Data Uncertainty (NGEU) for dimension reduction while incorporating the sampling distribution information. The method combines non-linear embeddings and data uncertainty with kernel mean embedding and optimize over a graph matching objective to preserve graph information. It also provides generalization guarantees in the form of Rademacher complexity. The Decision Recommendations given by three reviewers are respectively Leaning Reject, Learning Accept, and Reject. They all raised many concerns to address for the authors. Unfortunately, the authors did not provide a response to these concerns. Therefore, a negative decision is made. The authors are encouraged to significantly improve the work for other venues by considering the comments from the reviewers.

**Audience:**

Yes. There should be some researchers in this field interested in the work done by this paper. However, the findings in this paper are not solid enough to be publicly known in its current form. In addition, this paper might appeal to a small audience.

**Claims And Evidence:**

Some claims on contributions or strengths of the proposed method are not always convincingly supported by sufficient experiments, such as comparisons with other methods. It also suffers from unclear positioning in literature, such as distributional learning.